# IDENTIFIABLE REPRESENTATION LEARNING VIA ARCHITECTURE EQUIVARIANCES

## ABSTRACT

Despite their immense success and usefulness, current deep learning systems are still lacking in interpretability, robustness, and out of distribution generalisation. In this work we propose a method that helps address some of these issues in image and video data, by exploiting equivariances naturally present in the data. It enables learning latent representations that are identifiable and interpretable, and that can be intervened on to visualise counterfactual scenarios. The latent representations naturally correspond to positions of objects subject to image transformations, and so our method trains object detectors completely unsupervised, without object annotations. We prove that the learned latent variables are identifiable up to permutations and small shifts up to the size of model's receptive fields, and perform experiments demonstrating this in practice. We apply it to real world videos of balls moving in mini pool (translational equivariance), cars driving around a roundabout (rotational equivariance) and objects approaching the camera on a conveyor belt (scale equivariance). In all cases, transformation-equivariant representations are learned unsupervised. We show that intervening on the learned latent space results in successful generalisation out of the training distribution, and visualise realistic counterfactual videos never observed at training time. The method has natural applications in industry, such as inspection and surveillance, with static cameras.

## 1 INTRODUCTION

Some challenges facing current deep learning systems are interpretability, robustness, and out of distribution generalisation (Gilpin et al., 2018; Schölkopf et al., 2021). This is especially important in high-stakes domains such as healthcare and law, where for ML systems to be adopted, they need to be explainable, interpretable, and have guarantees about how they operate (Davenport & Kalakota, 2019; Bibal et al., 2021).

One way to address this problem is to exploit the knowledge of equivariances naturally present in the data. For example, in an object detection task, one might want to detect an object regardless of its position and so one might use a neural network architecture that is equivariant to translation. In another example, in a system for monitoring traffic at a roundabout, one might exploit the circular structure of the system and design an architecture that is equivariant to rotation. Or, if for example, one deals with egocentric footage of highway traffic where vehicles become smaller as they drive further, one might want to make use of equivariance to scale for recognising the vehicles. In all of these cases, making a network equivariant to the right transformations has multiple benefits, including making the latent space more interpretable, obtaining extra guarantees about the structure of the latent space, and better generalisation to unseen data that obey the same set of equivariances. Additionally, an equivariant network requires a smaller number of training samples as well as a smaller memory footprint due to weight sharing, thus reducing the time for data collection and training the network.

In this paper we propose a method to achieve this using an autoencoder-based architecture, where the encoder and decoder consist of blocks that make the latent representation equivariant to a specified transformation. This transformation is defined via a warping grid that can encode equivariances (e.g. to translations, rotations or scaling). The grid only needs to be specified once for each video scene, thus making it useful for inspection or surveillance applications, where cameras are typically static. Specifically, the encoder consists of a warping function followed by a standard CNN and a soft argmax function, and these operations are approximately inverted by the decoder. We prove that this configuration produces equivariant representations and also prove that the latent representation recovers the true variables (in this case, the objects' positions) up to small shifts. After training we

| Work | Generative | Any equivariance | Multiple objects | Identifiable |
|------|:----------:|:----------------:|:----------------:|:------------:|
| Ours | ✓ | ✓ | ✓ | ✓ |
| Henriques & Vedaldi (2017) | ✗ | ✓ | ✗ | ✗ |
| Jakab et al. (2018) | ✓ | ✗ | ✓ | ✗ |

Table 1: Comparison between the characteristics of our work and of two relevant related works.

can intervene on the latent variables and decode them into realistic counterfactual images and videos to visualise hypothetical scenarios never observed at training time.

Making networks equivariant to different transformations has been studied before (e.g. Cohen & Welling (2014); Sosnovik et al. (2020); Han et al. (2021), and others), however many works achieve this by focusing on the properties of kernels and on discrete transformations, while we focus on equivariance to continous transformations via input image warps. Equivariant and invariant networks were studied in different areas (Dieleman et al., 2015; Han et al., 2021; Lee et al., 2022; Pielawski et al., 2020; Musallam et al., 2022; Gupta et al., 2020), however most works focus on discriminative problems (classification or regression), while our focus is to generate counterfactual images and videos never seen at training time. Further, differently from previous works studying identifiability of neural networks (Hyvarinen & Morioka, 2016; 2017; Klindt et al., 2021; Khemakhem et al., 2020a;b; Zimmermann et al., 2021; Gresele et al., 2020), we obtain guarantees for the identifiability of the learned latent representation by imposing equivariances on the model architecture.

Concretely, our contributions in this paper are:

1. A novel generative, multi-object, equivariance-based method for learning latent representations of videos that are identifiable, interpretable, generalise out of the training distribution, and can be intervened on to generate counterfactual videos.

2. A proof of identifiability of the learned latent representation, showing that the latent variables are identifiable up to translations on the order of the model's receptive fields.

3. Various experiments demonstrating the method on real world videos, including balls moving in mini pool (translational equivariance), cars driving around a roundabout (rotational equivariance), and objects on a conveyor belt under perspective (scale equivariance). The experiments demonstrate identifiability in practice, as well as the ability to generate realistic counterfactual videos never seen at training time, by intervening on the learned latent space.

A direct comparison between our work and selected related works is shown in table 1.

## 2 RELATED WORK

Equivariances to different transformations in deep learning have been studied before. Cohen & Welling (2016) generalise CNNs to group equivariant CNNs (G-CNNs), however for many transformations this may require storing many filters. Gens & Domingos (2014) aims to achieve the same goal using Symmetry Networks. Cohen & Welling (2017) generalise G-CNNs to steerable CNNs which removes the memory scaling issue and allows working with infinite element groups. Cohen et al. (2019) propose gauge equivariant CNNs where the equivariance is to local gauge transformations on the surface of a sphere. Weiler & Cesa (2019) use E(2)-equivariant convolutions with steerable CNNs. Henriques & Vedaldi (2017) propose warped convolutions which achieve equivariance by warping the input image before passing it through a CNN. Focusing on specific transformations, Marcos et al. (2016; 2017); Li et al. (2018); Dieleman et al. (2015; 2016); Han et al. (2021); Pielawski et al. (2020); Gupta et al. (2020); Worrall et al. (2017) deal with equivariance and invariance to rotations and Kanazawa et al. (2014); Sosnovik et al. (2020) deal with equivariance and invariance to scale. In our work we deal with equivariances to continuous transformations (i.e. equivariance to a group with infinite number of elements), but we achieve this by warping the images, unlike for example steerable CNNs Cohen & Welling (2017) which achieves this using kernel properties. The closest work to ours is probably Henriques & Vedaldi (2017), however our method is generative while theirs is discriminative, and it has no guarantees of identifiability.

Equivariant networks have been applied to different areas. For example, Dieleman et al. (2015) use rotational invariance for galaxy classification, Han et al. (2021) use rotational equivariance for aerial object detection, Lee et al. (2022) use equivariance for keypoint detection in images, Pielawski et al. (2020) use rotational equivariance for image registration, Musallam et al. (2022) use equivariant

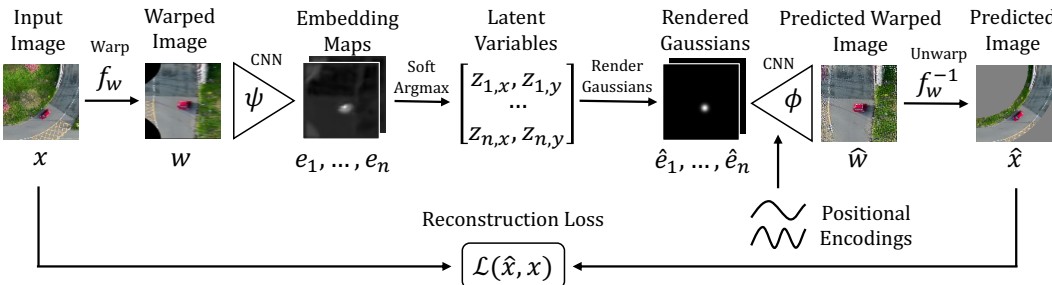

Figure 1: Network architecture, described from left to right. Encoder: (1) an image $x$ is warped using a function $f_w$ to obtain $w$, (2) it is then passed through a CNN $\psi$ to obtain $n$ embedding maps $e_1, ..., e_n$, (3) a maximum of each map is found using softargmax to obtain $[z_{1,x}, z_{1,y}, ..., z_{n,x}, z_{n,y}]$. Decoder: (1) Gaussians $\hat{e}_1, ..., \hat{e}_n$ are rendered at the positions given by the latent variables, (2) the Gaussian maps are concatenated with positional encodings and passed through a CNN $\phi$ to obtain the predicted warped image $\hat{w}$, (3) the image is unwarped by $f_w^{-1}$ to obtain the predicted image $\hat{x}$. Finally, $x$ and $\hat{x}$ are used to compute the reconstruction loss.

features for pose regresssion, and Gupta et al. (2020) use rotation equivariance for tracking. While most of the applications of equivariances have been discriminative (i.e. classification, regression), in this work we focus on generative modeling where we use equivariances to generate realistic data never observed at training time (counterfactuals).

Identifiability of learned representations has been studied in the field of causal representation learning Schölkopf et al. (2021). Locatello et al. (2019) have shown that learning identifiable latent variables is not possible in general without making assumptions about the model and the data. Thus, different works have made different assumptions about the distribution of the latent variables and about the mechanisms relating them (Hyvarinen & Morioka, 2016; 2017; Klindt et al., 2021; Khemakhem et al., 2020a;b; Zimmermann et al., 2021; Gresele et al., 2020); for an overview of identifiability assumptions in different works see Ahuja et al. (2022). Unlike previous works, we achieve identifiability by imposing grid-based spatial equivariances on the encoder and decoder architectures.

## 3 METHOD

In this section we present our method, which is based on an autoencoder architecture whose latent representation is equivariant to different transformations of the input images (fig. 1). We start with a brief discussion of translational equivariance in CNNs (sec. 3.1), followed by the description of the warping process we use to obtain different types of equivariances (sec. 3.2) and finally describing the representational bottleneck (sec. 3.3).

### 3.1 CNNS AND TRANSLATIONAL EQUIVARIANCE

Depending on the data, one might want to choose different parametrisations for the encoder and the decoder of an autoencoder. For example, without any prior knowledge one might parametrise $\psi$ and $\phi$ by MLPs, as they have been shown to be universal function approximators (Hornik et al., 1989). However, if one knows that e.g. translating an input image $x_t$ should result in a proportional shift in the latent variables $z_t$, one might choose to parametrise $\psi$ and $\phi$ by CNNs. This is referred to as translational equivariance and it can be generalised to a broader class of transformations such as rotations or scaling. In general, a network $\psi$ is equivariant to transformation $T$ if applying the transformation $T$ to the data before passing it through the network is equivalent to passing the data through the network and applying a transformation $T'$ afterwards, i.e.

$$\psi(T \circ x) = T' \circ \psi(x) \tag{1}$$

where $T$ and $T'$ may or may not be the same. CNNs consist of layers computing the convolution between a feature map $x$ and a filter $F$, defined in one dimension as

$$(x \star F)[i] = \sum_j x[j]F[j - i] \tag{2}$$

Intuitively, this corresponds to sliding the filter $F$ across the feature map $x$ and at each position of the filter $i$ computing the dot product between the feature map $x$ and the filter $F$. Convolutional layers

| Experiment | Inverse Warp | Forward Warp |
|---|---|---|
| Translation | $x = u_1, \;\; y = u_2$ | $u_1 = x, \;\; u_2 = y$ |
| Rotation | $x = a_1 + b_1 \cdot c^{u_1} \cos(u_2)$ $y = a_2 + b_2 \cdot c^{u_1} \sin(u_2)$ | $u_1 = \frac{1}{2} \log(c)^{-1} \log([\frac{x-a_1}{b_1}]^2 + [\frac{y-a_2}{b_2}]^2)$ $u_2 = \arctan_2(\frac{y-a_2}{b_2}, \frac{x-a_1}{b_1})$ |
| Scale | $x = a_1 + b_1 \cdot c_1^{u_1}$ $y = a_2 + b_2 \cdot c_2^{u_2}$ | $u_1 = \log(c_1)^{-1} \log(\frac{x-a_1}{b_1})$ $u_2 = \log(c_2)^{-1} \log(\frac{y-a_2}{b_2})$ |

Table 2: Summary of expressions used to perform forward and inverse warp for different experiments, expressed in terms of the original image coordinates $x, y$ and warped image coordinates $u_1, u_2$.

are equivariant to translations, i.e.

$$((\tau \circ x) \star F)[i] = \sum_j x[j - t] F[j - i] = \sum_j x[j] F[j - (i - t)] = \tau \circ (x \star F)[i] \qquad (3)$$

where $\tau$ is the translation operator that translates a feature map by $t$ pixels, and we have used the substitution $j \to j + t$ at the second equality. However, CNNs are not equivariant to other types of transformations such as rotations or scaling. We will now discuss one solution, using warping.

### 3.2 Generalised Equivariances via Warping

In order to achieve equivariance to a broader class of transformations, we can change the variables of the data from cartesian coordinates to a new set of coordinates that achieves the desired equivariance when shifted (similar to Henriques & Vedaldi (2017)). Formally, we define the *forward warp* $f_w$ as the invertible transformation that is applied to an image to change its coordinates to a new set of coordinates $(u_1, u_2)$ in which translation $\tau$ corresponds to the desired transformation $T$ in the original space (table 2, third column), and we define the *inverse warp* $f_w^{-1}$ as the inverse of this transformation (table 2, second column), i.e.

$$[f_w^{-1} \circ \tau \circ f_w](x) = T(x) \qquad (4)$$

For example, to obtain translational equivariance, $T = \tau$, one can set $f_w = I$ which means that the warped coordinates are identical to the original ones (table 2, first row; fig. 2, left column). To achieve equivariance to rotation transformations $T$, one can change the variables to polar coordinates using a polar warp $f_w$, where shifts along the angular dimension correspond to rotations in the original space (table 2, rows 2-3; fig. 2, middle column). Similarly, to achieve equivariance to scaling transformations $T$, one can use a logarithmic warping map $f_w$ to change the variables to log coordinates where the shifts correspond to scaling in the original space (table 2, rows 4-5; fig. 2, right column). Using this definition, we can prove that the warp $f_w$ post-composed with the encoder CNN $\psi$ is equivariant to the desired transformation $T$ on the input and to the translation $\tau$ on the output as

$$\psi \circ f_w(T \circ x) = \psi \circ f_w \circ (f_w^{-1} \circ \tau \circ f_w) \circ x = \psi \circ \tau \circ f_w \circ x = \tau \circ (\psi \circ f_w \circ x) \qquad (5)$$

where at the first equality we have used the definition of $T$ (eq. 4), at the second equality we have used the fact that $f_w \circ f_w^{-1} = I$ as $f_w$ is invertible, and at the third equality we have used the fact that the CNN $\psi$ is equivariant to translations $\tau$ (eq. 3). Note that Henriques & Vedaldi (2017) prove this equivariance only for exponential maps $f_w$, while our assumption is weaker, namely that $f_w$ has to be an invertible function that obeys $f_w^{-1} \circ \tau \circ f_w = T$ (eq. 4), or equivalently, $\tau \circ f_w = f_w \circ T$, thus generalising their proof.[1] We can prove a similar equivariance result for the decoder, namely that the decoder CNN $\phi$ post-composed with the inverse warp $f_w^{-1}$ is equivariant to the translation $\tau$ on the input and to the desired transformation $T$ on the output, i.e.

$$f_w^{-1} \circ \phi \circ (\tau \circ x) = f_w^{-1} \circ \tau \circ \phi \circ x = f_w^{-1} \circ \tau \circ (f_w \circ f_w^{-1}) \circ \phi \circ x = T \circ (f_w^{-1} \circ \phi \circ x) \qquad (6)$$

where at the first equality we have used the fact that the CNN $\phi$ is equivariant to translations $\tau$ (eq. 3), at the second equality we have inserted the identity $f_w \circ f_w^{-1} = I$, and at the third equality we

---

[1] For example, we can let $f_w$ be both a polar coordinate warp ($x = u_1 \cos u_2, y = u_1 \sin u_2$) and a log-polar coordinate warp ($x = e^{u_1} \cos u_2, y = e^{u_1} \sin u_2$), while the results of Henriques & Vedaldi (2017) only apply to the log-polar warp because it is an exponential map, and not to the standard polar warp.

**Translation**   **Rotation**   **Scale**

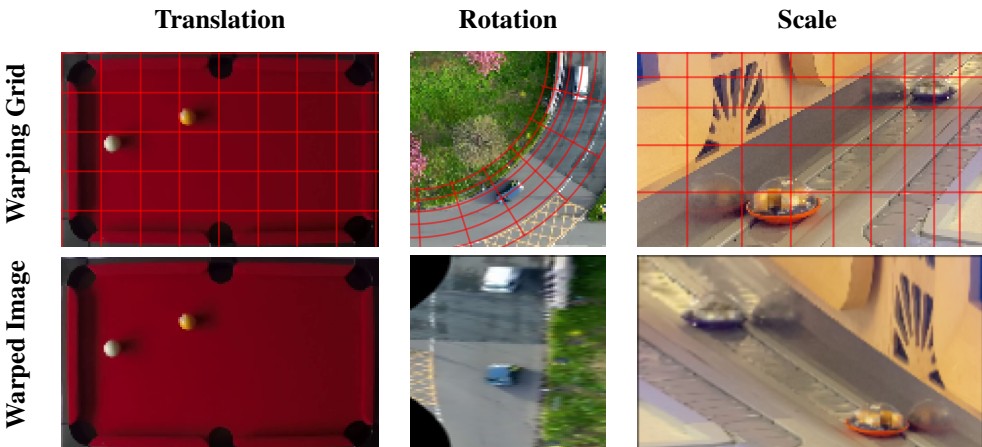

Figure 2: Warping grids overlayed over an image from each dataset (top row) and a corresponding warped image (bottom row) for translational, rotational and scale experiments (columns).

have used the definition of $T$ (eq. 4). In practice, we implement the forward and inverse warps $f_w$ and $f_w^{-1}$ by computing the *forward* and *inverse warping grids* $G_w$ and $G_w^{-1}$ offline by

$$G_w = \{f_w^{-1}(u_1, u_2) : (u_1, u_2) \in \{0, 1, ..., U_1\} \times \{0, 1, ..., U_2\}\} \tag{7}$$

$$G_w^{-1} = \{f_w(x, y) : (x, y) \in \{0, 1, ..., X\} \times \{0, 1, ..., Y\}\} \tag{8}$$

where $f_w$ and $f_w^{-1}$ are obtained from table 2 (columns 2-3), $X, Y$ are the image dimensions, and $U_1, U_2$ are the dimensions of the warped space (Henriques & Vedaldi, 2017). Note the correspondence of inverses between $G_w$ and $f_w^{-1}$, and between $G_w^{-1}$ and $f_w$. These grids are then used online to warp the images as $f_w(x) = x[G_w]$ and $f_w^{-1}(x) = x[G_w^{-1}]$ where $x[G]$ denotes sampling an image $x$ at the points defined by the grid $G$ using bilinear interpolation, which is a fast operation. Note that the warping grids only need to be defined once for every video scene, making it practical for applications where the camera is static. In the next section we discuss how these equivariances of feature maps relate to the learned latent representation.

### 3.3 FROM FEATURE MAPS TO VARIABLES

So far we have only worked with images and feature maps, but ultimately we would like to obtain scalar latent variables that are equivariant to transformations applied to the images. This is because dealing with scalars is more natural and interpretable than dealing with feature maps – for example, it is natural to think about an object's position in terms of its coordinates instead of a feature map. To do this, we first define a translation $\tau$ of a (1D) feature map $x$ and a translation $\tau'$ of a scalar $z$ as

$$\tau(x)[i] = x[i - t], \quad \tau'(z) = z + t \tag{9}$$

where $i$ is the position in the feature map $x$, $\tau$ shifts an image by $t$ pixels, and $\tau'$ shifts a scalar by $t$ units. To relate translations in feature maps to translations in latent variables, we can use a function that computes a scalar property of a feature map $x$, such as argmax, defined as $\mathrm{argmax}(x) = \{i : x[j] \leq x[i] \ \forall j\}$. Using these definitions we can now prove the equivariance of argmax, i.e. that shifting the feature map $x$ by $\tau$ corresponds to shifting the latent variable $\mathrm{argmax}(x)$ by $\tau'$:

$$\mathrm{argmax}(\tau \circ x) = \{i : \tau \circ x[j] \leq \tau \circ x[i] \ \forall j\} = \{i : x[j - t] \leq x[i - t] \ \forall j\}$$
$$= \{i + t : x[j] \leq x[i] \ \forall j\} = \mathrm{argmax}(x) + t = \tau' \circ \mathrm{argmax}(x) \tag{10}$$

where at the first equality we have used the definition of argmax, at the second equality we have used the definition of $\tau$ (eq. 9, left), at the third equality we have used the substitution $i \to i + t$, at the fourth equality we have used the definition of argmax, and at the last equality we have used the definition of $\tau'$ (eq. 9, right). Similarly, to now relate shifts in latent variables $z$ to shifts of feature maps $x$, we can invert the action of the argmax operation. Because argmax is a many-to-one function, finding an exact inverse is not possible, but we can obtain a pseudo-inverse using the delta function defined as $\mathrm{delta}(z)[i] = \delta(i - z)$ where $\delta$ is the Dirac delta function. We can show that delta is a pseudo-inverse of argmax because $\mathrm{argmax} \circ \mathrm{delta} \circ z = \{i : \delta(x - z)[j] \leq \delta(x - z)[i] \ \forall j\} = z$.

Now, similar to the $\mathrm{argmax}$ function, we can prove that the $\mathrm{delta}$ function is equivariant to the latent variable shift $\tau'$ on the input and the feature map shift $\tau$ on the output, i.e.

$$\mathrm{delta}(\tau' \circ z)[i] = \delta(i - \tau' \circ z) = \delta(i - z - t) = \mathrm{delta}(z)[i - t] = \tau \circ \mathrm{delta}(z)[i] \quad (11)$$

where at the first equality we have used the definition of $\mathrm{delta}$, at the second equality we have used the definition of $\tau'$ (eq. 9, right), at the third equality we have used the definition of $\mathrm{delta}$, and at the last equality we have used the definition of $\tau$ (eq. 9, left). Now we have the tools to convert between equivariances in feature maps and latent variables via the functions $\mathrm{argmax}$ and $\mathrm{delta}$. However, because these operations are not differentiable, for neural network training we approximate $\mathrm{argmax}$ via a differentiable function $\mathrm{softargmax}$, defined in two dimensions as

$$\mathrm{softargmax}(x) = \left( \frac{1}{I} \sum_{i=0}^{I} \sum_{j=0}^{J} i \, \sigma_1 \left( \frac{x}{\Theta} \right)[i,j], \; \frac{1}{J} \sum_{i=0}^{I} \sum_{j=0}^{J} j \, \sigma_2 \left( \frac{x}{\Theta} \right)[i,j] \right) \quad (12)$$

where $\sigma$ is the softmax function defined in one dimension as $\sigma(x)[i] = \exp(x[i])/\sum_j \exp(x[j])$, $\sigma_1(x)$ and $\sigma_2(x)$ is the softmax function evaluated along the first and second dimensions of $x$, $\Theta$ is a temperature hyperparameter, $[i,j]$ is the image index, $I$ is the image width, and $J$ is the image height. As the temperature $\Theta$ in 12 approaches zero, $\mathrm{softargmax}$ reduces to the classical $\mathrm{argmax}$ function. Similarly, we can approximate the hard $\mathrm{delta}$ function using a differentiable $\mathrm{render}$ function as

$$\mathrm{render}(z)[i] = \mathcal{N}(i - z, \sigma^2) \quad (13)$$

where $\mathcal{N}(i - z, \sigma^2)$ is a normal distribution evaluated at $i - z$ with variance given by the hyperparameter $\sigma^2$. As the variance $\sigma^2$ in eq. 13 approaches zero, the $\mathrm{render}$ function reduces to the hard $\mathrm{delta}$ function. Therefore, now we have all the elements we need to create an equivariant architecture where the encoder and decoder are defined, respectively, by

$$z_t = \mathrm{softargmax} \circ \psi \circ f_w \circ x_t, \quad \hat{x}_t = f_w^{-1} \circ \phi \circ \mathrm{render} \circ z_t. \quad (14)$$

This is illustrated in fig. 1. In the next section we prove identifiability of the learned latent variables.

## 4 THEORETICAL RESULTS

In this section we show that the learned latent variables are identifiable with respect to the ground truth physical variables, up to permutations and small shifts.

**Theorem 1** (Identifiability of latent representation). *Consider an image $x_t$ with objects of size $s_O$, warping map $f_w$, CNN encoder $\psi$ with receptive field size $s_\psi$, CNN decoder $\phi$ with receptive field size $s_\phi$, soft argmax function* $\mathrm{softargmax}$*, Gaussian rendering function* $\mathrm{render}$*, and latent variables $z_t$, composed as $z_t = \mathrm{softargmax} \circ \psi \circ f_w \circ x_t$ and $\hat{x}_t = f_w^{-1} \circ \phi \circ \mathrm{render} \circ z_t$ (fig. 1). Assuming*

*(A1) The reconstruction loss is minimised,* $\min_{\psi,\phi} \mathcal{L}(\hat{x}, x)$*.*

*(A2) Each object has at least two distinct positions in the training set.*

*(A3) The warping map $f_w$ is a diffeomorphism.*

*(A4) There are no two identical objects in any image $x_t$.*

*(A5) Each image $x_t$ has the same background.*

*(A6) The Gaussian rendered by the* $\mathrm{render}$ *function is a delta function.*

*Then the latent variables $z_t$ are identified up to permutations and maximum shifts of $\min(s_\psi + f_w(s_O), s_\phi)/2$. For the special case that $s_\psi = s_\phi = s_{RF}$, the shifts reduce to $s_{RF}/2$.*

Here we present a proof sketch; for a full proof see Appendix A. First, minimising the reconstruction loss (A1) means that the objects in the predicted image have to be reconstructed at the same positions as in the original image. Then, the dataset having each object present at a minimum of 2 different positions (A2) ensures that the latent variables used by the decoder must contain information about each object, and thus the encoder must learn to match all objects. Next, the warp being a diffeomorphism (A3), the encoder being equivariant to the transformation that generated the data (eq. 5), and each image containing distinct objects (A4) on a static background (A5) ensure that each different object is mapped to a unique latent variable. This variable is correct up to a small shift, because any part of the receptive field of the encoder can match any part of the (warped) object, $(s_\psi + f_w(s_O))/2$, not just the center. Similarly, when decoding there is possibly another small shift because any part of the decoder filter may be convolved with the rendered delta function (A6), i.e. $s_\phi/2$. Because the

| | MLP | | CNN | | Keypoint CNN | | Proposed Method | |
|---|---|---|---|---|---|---|---|---|
| | MSE | Acc. | MSE | Acc. | MSE | Acc. | MSE | Acc. |
| Translation | 2.05 | 98.3% | $1.6 \cdot 10^5$ | 99.0% | – | – | $\mathbf{8.2 \cdot 10^{-3}}$ | **99.6**% |
| Rotation | 1.92 | **97.3**% | $9.6 \cdot 10^3$ | 95.5% | 0.197 | 96.9% | $\mathbf{1.7 \cdot 10^{-2}}$ | **97.3**% |
| Scale | 7.25 | 96.9% | $4.3 \cdot 10^4$ | 92.2% | 0.192 | 97.1% | $\mathbf{1.9 \cdot 10^{-2}}$ | **97.5**% |

Table 3: Results showing the mean squared error of the predicted latent variables w.r.t. estimated ground truth physical variables (MSE, lower is better) and the image reconstruction accuracy of the decoded video frames w.r.t. input video frames (Acc., higher is better). Results are reported for the proposed method and for MLP, CNN, and keypoint CNN baselines for each experiment.

predicted and original objects must have the same position (A1), the shifts from the encoder and the decoder have to cancel each other, and thus the latent variables are shifted by a maximum amount of $\min((s_\psi + f_w(s_O))/2, s_\phi/2)$. Additionally, because the objects can be mapped to the variables in an arbitrary order, there is additional non-identifiability due to object permutations.

## 5 EXPERIMENTS

In this section we present 3 experiments validating our method from sec. 3: one using translational equivariance (sec. 5.1), one using rotational equivariance (sec. 5.2), and one using scale equivariance (sec. 5.3). In each experiment we demonstrate that making the network architecture equivariant to a transformation naturally present in each dataset allows one to identifiably learn latent variables corresponding to the ground truth physical variables (table 3, MSE), and to intervene on the learned latent variables (fig. 3) to generate realistic counterfactual videos never seen at training time (fig. 4). We implement the method described in sec. 3 using the architecture in fig. 1, with the warps summarised in table 2. For comparison, in each experiment we also train 3 analogous baseline models: MLP, CNN and keypoint CNN (Jakab et al., 2018). For implementation details see appendix B.

### 5.1 TRANSLATION

**Setup.** The training and test sets for this experiment consist of 15 and 11 frames respectively from a video of two balls moving on a mini pool table, visualised in fig. 4, upper left plot. Because the table naturally extends horizontally and vertically, we seek to employ an autoencoder architecture that is equivariant to horizontal and vertical translations. Because a standard CNN is already translationally equivariant, we use a standard CNN encoder and decoder with an identity warp (table 2, first row) visualised in fig. 2 (first column).

**Identifiability results.** The latent variables corresponding to the training data are visualised in fig. 3 (left plot) in blue and purple for the first and second balls respectively, resulting in straight lines for the moving balls as expected. When compared to the estimated ground truth variables describing the balls' position, the latent variables mean squared error on the test set is very small, orders of magnitude smaller than the baselines (table 3, MSE, top row). This is to be expected as an MLP architecture does not exhibit any equivariances and so performs poorly on the test set, where the balls are now at positions never encountered at training time. Similarly, the CNN baseline achieves a comparably poor performance, because while the network contains convolutional layers, the translational equivariance is broken by the linear layer mapping features to latent variables. In this case, the keypoint CNN baseline is equivalent to our method due to the warp being an identity, and so is not included in table 3. Our method also achieves the best test set reconstruction accuracy (table 3, Acc., top row) as the translational equivariance allows it to successfully generalise to the test set.

**Counterfactual results.** Once the mapping between the images and the latent space has been learned, we can use the translational equivariance property of the network to intervene on the latent variables and generate videos of counterfactual scenarios that were never observed at training time. For example, one can visualise the balls moving in opposite directions at a constant speed (fig. 3, left plot, red and orange; fig. 4, upper middle plot) and the white ball bouncing off one of the table edges while slowing down (fig. 3, left plot, green; fig. 4, upper right plot). Note that none of these scenarios were observed at training time, demonstrating that the model successfully generalises out of the training distribution. It also allows controlled generation with interpretable latent variables.

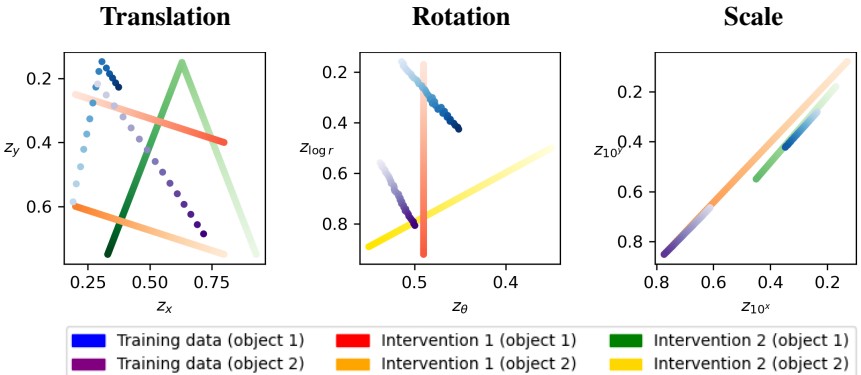

Figure 3: Latent space showing the training data and two different counterfactuals for each experiment that are out of the training distribution. $z_x$ and $z_y$ denotes the horizontal and vertical position, $z_\theta$ and $z_{\log r}$ are the angular and (log) radial position, and $z_{10^x}$ and $z_{10^y}$ are the horizontal and vertical position on a log scale. The colour intensity denotes the arrow of time (light to dark).

## 5.2 ROTATION

**Setup.** The training and test sets consist of 35 and 15 frames each from a video of two cars driving around a quarter of a roundabout, visualised in fig. 4, middle left plot. Because the cars at the roundabout can move in an angular (forward or backward) or radial (change lanes) direction, we would like to employ an autoencoder architecture which is equivariant to rotation and radial shifts around the center of the roundabout. We achieve this by using a log-polar warp (table 2, middle row), visualised in fig. 2 (second column), together with a standard CNN autoencoder.

**Identifiability results.** The latent variables corresponding to the training data are visualised in fig. 3, middle plot, in blue and purple for the two cars respectively. The data forms two approximately straight lines with a steadily increasing angular position (and a slightly increasing radial position), as expected. When compared to the estimated ground truth variables describing the car's position, the latent variables' mean squared error on the test set is an order of magnitude smaller than the best baseline (table 3, MSE, middle row), which reflects the fact that none of the baselines exhibit equivariance to rotation and radial position. Consequently, our method also achieves the best test set reconstruction accuracy (table 3, Acc., middle row) as the rotational equivariance allows it to generalise successfully to the test set. Although the MLP baseline achieves a comparable reconstruction accuracy to our method, this is misleading because the MLP renders the objects at incorrect positions and the high accuracy arises due to better reconstruction of the background, whereas our method reconstructs the cars at the correct positions albeit with slightly more noise.

**Counterfactual results.** Once the encoder and decoder have been learned, we can use the rotational and radial equivariance property of the network to intervene on the latent variables and generate videos of counterfactual scenarios that were never observed at training time. For example, one can make the first latent variable have a constant radial distance and an increasing angular position (fig. 3, middle plot, red) to visualise the white car continuing to drive around the whole roundabout (fig. 4, center plot), or have the second variable increase its angular position and decrease its radial position (fig. 3, middle plot, yellow) to visualise the blue car moving forward while changing lanes at the same time (fig. 4, top right plot). Because none of these scenarios were observed at training time, this demonstrates that the model successfully generalises out of the training distribution.

## 5.3 SCALE

**Setup.** The training and test sets consist of 79 and 40 frames respectively from a video of two sushi bowls moving closer to the camera on a conveyor belt (visualised in fig. 4, bottom left plot). Because the bowls have a different scale depending on their position, we would like to employ an autoencoder architecture that is equivariant to scale. We achieve this by using a scale warp (table 2, bottom row) visualised in fig. 2 (right column), together with a standard CNN autoencoder.

**Identifiability results.** The latent variables corresponding to the training data are visualised in fig. 3 (right plot, blue and purple for the two bowls respectively). The data forms a diagonal line in the latent space in logarithmic coordinates, showing an approximately exponential relationship between

**Training Data**                    **Counterfactuals**

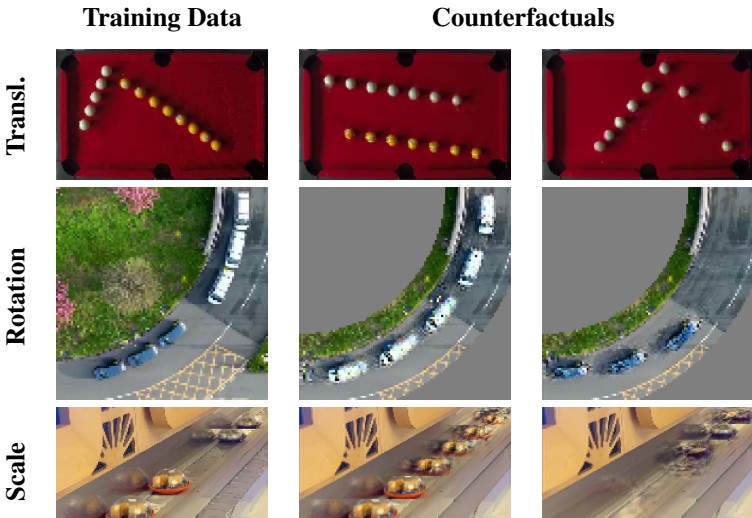

Figure 4: Training data (left column) and two different counterfactuals (middle and right columns) for each experiment (rows). Our method learns to detect the objects unsupervised, with no object annotations. Additionally, it can generate images and extrapolate them to new situations. The counterfactuals are obtained by intervening on and decoding the latent variables to obtain out-of-distribution data never seen during training.

the bowls' position and scale. When compared to the estimated ground truth variables describing the bowls' position, the latent variables' mean squared error on the test set is an order of magnitude smaller than the best baseline (table 3, MSE, bottom row), which is to be expected as none of the baselines exhibit scale equivariance. Our method also achieves the best reconstruction accuracy (table 3, Acc., bottom row) as the scale equivariance allows it to successfully generalise to the test set.

**Counterfactual results.** Once the encoder and decoder have been learned, we can use the scale equivariance property of the network to intervene on the latent variables and generate videos of counterfactual scenarios that were never observed at training time. For example, one can extrapolate the latent variables for the first object (fig. 3, right plot, orange) to visualise where the orange bowl has been in the past (fig. 4, bottom middle plot), or extrapolate the variables for the second object in both directions (fig. 3, right plot, green) to visualise where the blue bowl was in the past and where it will be in the future, assuming constant speed (fig. 4, bottom right plot). Because none of these scenarios were observed at training time this demonstrates the model successfully generalising out of the training distribution. We note that it is naturally easier to extrapolate from larger to smaller scales (orange bowl, fig. 4, bottom middle plot) than it is to extrapolate in the opposite direction (blue bowl, fig. 4, bottom right plot) since more details are required to extrapolate to larger than smaller scales.

## 6    CONCLUSION

In this work we presented a method for learning an identifiable and interpretable latent representation of images and videos by exploiting equivariances naturally present in the data. We achieved this using an autoencoder architecture where the image is warped by a map corresponding to a specified equivariance before being passed through a CNN and a softargmax operation, and it is reconstructed by inverting this process. We proved that the learned latent representation is identifiable with respect to the ground truth variables and demonstrated this experimentally. We then applied the method to real world videos with multiple objects and different naturally present equivariances, and showed that by intervening on the latent representation we can generate realistic counterfactual videos that were never observed at training time. It also works as an unsupervised object detector, trained using raw video footage. In future work we would like to expand the current class of equivariance transformations and consider dealing with non-static backgrounds.

**Reproducibility Statement: we will make all source code available upon publication.**

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

## A  PROOF OF IDENTIFIABILITY

Here we present a proof of Theorem 1, which we reproduce for ease of reference.

**Theorem 2** (Identifiability of latent representation). *Consider an image $x_t$ with objects of size $s_O$, warping map $f_w$, CNN encoder $\psi$ with receptive field size $s_\psi$, CNN decoder $\phi$ with receptive field size $s_\phi$, soft argmax function* softargmax, *Gaussian rendering function* render, *and latent variables $z_t$, composed as $z_t = \text{softargmax} \circ \psi \circ f_w \circ x_t$ and $\hat{x}_t = f_w^{-1} \circ \phi \circ \text{render} \circ z_t$ (fig. 1). Assuming*

*(A1)  The reconstruction loss is minimised, $\min_{\psi,\phi} \mathcal{L}(\hat{x}, x)$.*

*(A2)  Each object has at least two distinct positions in the training set.*

*(A3)  The warping map $f_w$ is a diffeomorphism.*

*(A4)  There are no two identical objects in any image $x_t$.*

*(A5)  Each image $x_t$ has the same background.*

*(A6)  The Gaussian rendered by the* render *function is a delta function.*

**Encoder Uncertainty**

Encoder Filter

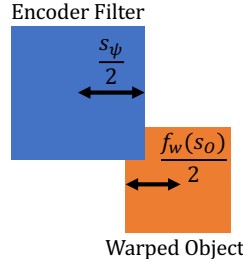

**Decoder Uncertainty**

Decoder Filter

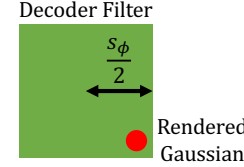

Rendered Gaussian

Warped Object

Figure 5: Left: uncertainty in object position due to any part of the encoder filter (blue) matching any part of the warped object (orange), with a maximum magnitude of $(s_\psi + f_w(s_O))/2$. Right: uncertainty of the rendered image position due to any part of the decoder filter (green) matching any part of the rendered Gaussian (red), with a maximum magnitude of $s_\psi/2$.

*Then the latent variables $z_t$ are identified up to permutations and maximum shifts of* $\min(s_\psi + f_w(s_O), s_\phi)/2$. *For the special case that $s_\psi = s_\phi = s_{RF}$, the shifts reduce to $s_{RF}/2$.*

*Proof.* By assumption (A1), the reconstruction loss $\mathcal{L}(\hat{x}_t, x_t)$ being minimised results in $\hat{x}_t = x_t$. This is valid for any loss function that is differentiable and whose minimum occurs when the reconstructed image is the same as the original image, i.e. when $\min_{\hat{x}_t} \mathcal{L}(\hat{x}_t, x_t) = x_t$ (for example, the mean squared error loss function or the binary cross-entropy loss function). Therefore, the positions of the objects in the original image have to be the same as the positions of the objects in the reconstructed image.

By assumption (A2) (each object appears at minimum 2 different positions), the latent variables used by the decoder have to contain some information about each object, and thus the encoder has to learn to match all the objects. This is because the decoder CNN $\phi$ takes as its input the rendered Gaussians $\hat{e}_t = \text{render} \circ z_t$ concatenated with positional encodings (fig. 1), and if some object in the dataset only appeared at a single position the model could achieve perfect reconstruction solely by using the positional encodings without having to use the Gaussian maps $\hat{e}_t$. However, because the dataset contains each object at minimum 2 positions, relying purely on positional encodings is now not sufficient, as without any information about the object passed to the decoder, the decoder wouldn't be able to know where to render the object. Formally, because $\hat{x}_t = f_w^{-1} \circ \phi \circ \text{render} \circ \text{softargmax} \circ \psi \circ f_w \circ x_t$, this means that for $\hat{x}_t = x$ (A1), the encoder $\psi$ needs to learn to match some part of each object in $x_t$.

Next, assumption (A3) (the warping function $f_w$ being a diffeomorphism) means that there is a one-to-one mapping between the original image $x_t$ and the warped image $w_t$, so that an original image $x_t$ with an object at position $(x, y)$ is mapped to a warped image $w_t = f_w \circ x_t$ with an object at position $(u_1, u_2) = f_w(x, y)$, where $f_w$ is the forward warp (sec. 3.2).

Because $z_t = \text{softargmax} \circ \psi \circ w_t$ is equivariant to translations of $w_t$ (sec. 3.1, 3.3), and because the encoder $\psi$ has to match some part of each object in $w_t$ (as stated previously), and also each image consists of distinct objects (A4) on a static background (A5), the warped image $w_t$ with an object at the position $(u_1, u_2)$ is encoded by $\text{softargmax} \circ \psi$ to the latent variables

$$z_t = (u_1 + \Delta_{\psi 1}, u_2 + \Delta_{\psi 2}), \quad |\Delta_{\psi 1}|, |\Delta_{\psi 2}| \leq \frac{s_\psi}{2} + \frac{f_w(s_O)}{2} \tag{15}$$

where the shifts $\Delta_{\psi 1}$ and $\Delta_{\psi 2}$ arise because any part of the encoder filter (of size $s_\psi$) can match any part of the object in the warped image (of size $f_w(s_O)$). See fig. 5 (left) for an illustration.

Next, because $\hat{w}_t = \phi \circ \text{render} \circ z_t$ is equivariant to translations of $z_t$, the latent variables $z_t = (u_1 + \Delta_{\psi 1}, u_2 + \Delta_{\psi 2})$ are mapped to a predicted warped image $\hat{w}_t = \phi \circ \text{render} \circ z_t$ with an object at position

$$(u_1 + \Delta_{\psi 1} + \Delta_{\phi 1}, u_2 + \Delta_{\psi 2} + \Delta_{\phi 2}), \quad |\Delta_{\phi 1}|, |\Delta_{\phi 2}| \leq \frac{s_\phi}{2} \tag{16}$$

where the shifts $\Delta_{\phi 1}$ and $\Delta_{\phi 2}$ arise because any part of the decoder filter (of size $s_\phi$) can match the rendered Gaussian $\hat{e}_t = \text{render} \circ z_t$, assumed to be a delta function (A6). See fig. 5 (right) for illustration.

Since, by assumption (A3), $f_w$ is a diffeomorphism, there is a one-to-one mapping from the predicted warped image $\hat{w}_t$ to the predicted image $\hat{x}_t = f_w^{-1} \circ \hat{w}_t$. Therefore, the predicted warped image $\hat{w}_t$ with an object at position $(u_1 + \Delta_{\psi 1} + \Delta_{\phi 1}, u_2 + \Delta_{\psi 2} + \Delta_{\phi 2})$ is mapped to a predicted image $\hat{x}_t = f_w^{-1} \circ \hat{w}_t$ with an object at the position

$$f_w^{-1}(u_1 + \Delta_{\psi 1} + \Delta_{\phi 1}, u_2 + \Delta_{\psi 2} + \Delta_{\phi 2}) \tag{17}$$

Finally, by assumption (A1) ($\hat{x}_t = x_t$), the position of each object in the original image $x_t$ has to be equal to the position of the object in the reconstructed image, i.e.

$$(x, y) = f_w^{-1}(u_1 + \Delta_{\psi 1} + \Delta_{\phi 1}, u_2 + \Delta_{\psi 2} + \Delta_{\phi 2}) \tag{18}$$

Applying $f_w$ from the left to both sides of the equation and using $f_w(x, y) = (u_1, u_2)$ and $f_w \circ f_w^{-1} = I$ (A3) results in the conditions

$$\Delta_{\psi 1} + \Delta_{\phi 1} = 0, \quad \Delta_{\psi 2} + \Delta_{\phi 2} = 0 \tag{19}$$

and therefore

$$|\Delta_{\psi 1}| = |\Delta_{\phi 1}|, \quad |\Delta_{\psi 2}| = |\Delta_{\phi 2}| \tag{20}$$

In words, this means that the shift in the latent variables acquired from the encoder $\Delta_\psi$ has to be balanced by an opposite the shift of the same magnitude in the decoder $\Delta_\phi$ in order to reconstruct the object at the same position. Because the shift due to the encoder is of maximum magnitude of $(s_\psi + f_w(s_O))/2$ and the shift due to the decoder is of maximum magnitude of $s_\phi/2$, this means that the maximum magnitude of the shift of the latent variables has to be the minimum of these two expressions, i.e. the learned latent variables $(u_1, u_2)$ are identifiable with respect to the ground truth latent variables $(u_1^{GT}, u_2^{GT})$ up to

$$(u_1, u_2) = (u_1^{GT} + \Delta_1, u_2^{GT} + \Delta_2), \quad |\Delta_1|, |\Delta_2| \leq \frac{\min(s_\psi + f_w(s_O), s_\phi)}{2} \tag{21}$$

For the common case that the encoder and decoder receptive fields are the same, i.e. $s_\psi = s_\phi = s_{RF}$, this simplifies to

$$(u_1, u_2) = (u_1^{GT} + \Delta_1, u_2^{GT} + \Delta_2), \quad |\Delta_1|, |\Delta_2| \leq \frac{s_{RF}}{2} \tag{22}$$

Additionally, because the order in which the objects get mapped to each latent variable is arbitrary, there is an additional non-identifiability arising due to variable permutations (for example, for two objects, $z_1$ can correspond to the first object's position and $z_2$ to the second object's position, or vice versa). □

# B IMPLEMENTATION DETAILS

## B.1 MAIN EXPERIMENTS

We implement the method described in sec. 3 using the architecture in fig. 1. The forward and inverse warps $f_w$ and $f_w^{-1}$ are performed according to the expressions in table 2. The encoder $\psi$ and decoder $\phi$ are both 5-layer convolutional neural networks (CNNs) with 32 channels, kernel size 7, stride 1, and padding 3, and with Batch Normalization Ioffe & Szegedy (2015) and ReLU activations between each layer. Additionally, the first convolutional layer of the encoder has 3 channels and the last one has $n$ (where $n$ is the number of variables), and the first convolutional layer of the decoder has 32+$n$ channels and the last one has 3. The positional encodings have 32 channels and have the dimensions of the images in each dataset. The reconstruction loss is a binary cross-entropy loss with logits applied to the input image $x_t$ and the predicted image $\hat{x}_t$ which are masked such that they only contain pixels that are inside the warping grid. We optimise the reconstruction loss using the Adam optimiser Kingma & Ba (2015) with learning rates $\alpha \in \{10^{-3}, 3 \cdot 10^{-4}\}$ and batch size 128 until convergence, and select the run with the best test set reconstruction accuracy. Image dimensions are $3 \times 100 \times 163$ px for the translation experiment (sec. 5.1), $3 \times 100 \times 100$ px for the rotation experiment (sec. 5.2), and $3 \times 100 \times 177$ px for the scale experiment (sec. 5.3).

| | MLP | | CNN | | Keypoint CNN | | Proposed Method | |
|---|---|---|---|---|---|---|---|---|
| | MSE | Acc. | MSE | Acc. | MSE | Acc. | MSE | Acc. |
| Translation | 46.2 | 97.2% | 1.06 | 92.8% | – | – | $\mathbf{7.2 \cdot 10^{-5}}$ | **98.6%** |
| Rotation | 4.02 | 96.9% | 2.59 | 85.6% | 0.489 | 96.5% | $\mathbf{3.4 \cdot 10^{-3}}$ | **97.6%** |
| Scale | 36.7 | 98.7% | $1.0 \cdot 10^{10}$ | **99.3%** | 0.112 | **99.3%** | $\mathbf{1.5 \cdot 10^{-2}}$ | 98.9% |

Table 4: Results showing the mean squared error of the predicted latent variables w.r.t. estimated ground truth physical variables (MSE, lower is better) and the image reconstruction accuracy of the decoded video frames w.r.t. input video frames (Acc., higher is better). Results are reported for the proposed method and for MLP, CNN, and keypoint CNN baselines for each experiment.

## B.2 BASELINES

For each experiment we also train 3 baseline models: MLP, CNN and keypoint CNN. The MLP baseline is an autoencoder where the encoder and decoder are both 5-layer MLPs with 32 hidden units, and with Batch Normalization Ioffe & Szegedy (2015) and ReLU activations between each layer. The CNN baseline is an autoencoder where the encoder and decoder are both 5-layer CNNs analogous to our model but the last convolutional layer of the encoder has 3 channels and is followed by Batch Normalization, ReLU, and a linear layer, and the decoder is a linear layer followed by Batch Normalization, ReLU, and an identical CNN, where the first convolutional layer has 3 channels. Finally, the keypoint CNN baseline (similar to Jakab et al. (2018)) is equivalent to our model but without warping.

## C ADDITIONAL EXPERIMENTS

In this section we present 3 additional experiments validating our method from sec. 3: one using translational equivariance (sec. C.1), one using rotational equivariance (sec. C.2), and one using scale equivariance (sec. C.3). In each experiment we demonstrate that making the network architecture equivariant to a transformation naturally present in each dataset allows one to identifiably learn latent variables corresponding to the ground truth physical variables (table 4, MSE) and to intervene on the learned latent variables (fig. 7) to generate realistic counterfactual videos never seen at training time (fig. 8).

### C.1 TRANSLATION

**Setup.** The training and test set for this experiment consists of 25 frames each from a video of a car moving for a short distance in a single lane, visualised in fig. 8, upper left plot. Because the car can move forward or backward and also change lanes (left or right), we seek to employ an encoder-decoder architecture which is equivariant to horizontal and vertical translations. Because a standard CNN is already translationally equivariant, we use a standard CNN encoder and decoder with an identity warp (table 2, first row) visualised in fig. 6 (first column).

**Identifiability results.** The latent variables corresponding to the training and test data are visualised in fig. 7 (left plot) in blue and red respectively, resulting in an approximately straight line as expected. When compared to the estimated ground truth variables describing the car's position, the latent variables mean squared error on the test set is very small, 5 orders of magnitude smaller than the baselines (table 4, MSE, top row). This is to be expected as an MLP architecture does not exhibit any equivariances and so performs poorly on the test set where the car is now at a position never encountered at training time. Similarly, the CNN baseline achieves a comparably poor performance because while the network contains convolutional layers, the translational equivariance of the former is broken by the linear layer at the end of the encoder. In this case the keypoint CNN baseline is equivalent to our method due to the warp being an identity and so is not included in table 4. Our method also achieves the best test set reconstruction accuracy (table 4, Acc., top row) as the translational equivariance allows it to successfully generalise to the test set.

**Counterfactual results.** Once the mapping between the images and the latent space has been learned, we can use the translational equivariance property of the network to intervene on the latent variables and generate videos of counterfactual scenarios that were never observed at training time. For example, one can visualise the car driving in a different lane (fig. 7, left plot, orange; fig. 8, upper middle plot) and the car accelerating while changing lanes (fig. 7, left plot, green; fig. 8, upper right

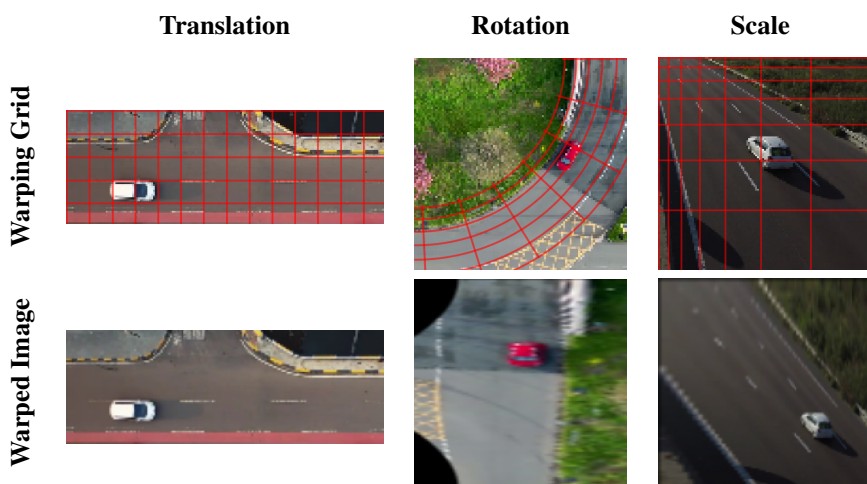

Figure 6: Warping grids overlayed over an image from each dataset (top row) and a corresponding warped image (bottom row) for translational, rotational and scale experiments (columns).

plot). Note that none of these scenarios were observed at training time, demonstrating that the model successfully generalises out of the training distribution.

## C.2 ROTATION

**Setup.** The training and test sets consist of 40 frames each from a video of a car driving around a small portion of a roundabout, visualised in fig. 8, middle left plot. Because the car at the roundabout can move in an angular (forward or backward) or radial (change lanes) direction, we would like to employ an encoder-decoder architecture which is equivariant to rotation and radial shifts around the center of the roundabout. We achieve this by using a log-polar warp (table 2, middle row), visualised in fig. 6 (second column), together with a standard CNN encoder/decoder.

**Identifiability results.** The latent variables corresponding to the training and test data are visualised in fig. 7, middle plot, in blue and red respectively. The data forms an approximately straight line with a constant radial distance and a steadily increasing angular position, as expected. When compared to the estimated ground truth variables describing the car's position, the latent variables mean squared error on the test set is 2 orders of magnitude smaller than the best baseline (table 4, MSE, middle row), which is to be expected as none of the baselines exhibit equivariance to rotation and radial position. Consequently, our method also achieves the best test set reconstruction accuracy (table 4, Acc., middle row) as the rotational equivariance allows it to generalise successfully to the test set.

**Counterfactual results.** Once the encoder and decoder have been learned, we can use the rotational and radial equivariance property of the network to intervene on the latent variables and generate videos of counterfactual scenarios that were never observed at training time. For example, one can make the variables to have a greater constant radial distance and an increasing angular position (fig. 7, left plot, orange colour) to visualise the car driving in a different lane (fig. 8, center plot), or have a non-linear relationship between the radial and angular positions (fig. 7, left plot, green colour) to visualise a car accelerating while changing lanes at the same time (fig. 8, middle right plot). Because none of these scenarios were observed at training time, this demonstrates that the model successfully generalises out of the training distribution.

## C.3 SCALE

**Setup.** The training and test sets consist of 13 and 27 frames from a video of a car driving on a highway and becoming visually smaller the further it is (visualised in fig. 8, bottom left plot). Because the car has a different scale depending on its position, we would like to employ an encoder-decoder architecture which is equivariant to scale. We achieve this by using a scale warp (table 2, bottom row) visualised in fig. 6 (right column), together with a standard CNN encoder-decoder.

**Identifiability results.** The latent variables corresponding to the training and test data are visualised in fig. 7 (middle plot, blue and red colours respectively). The data forms a diagonal line in the latent

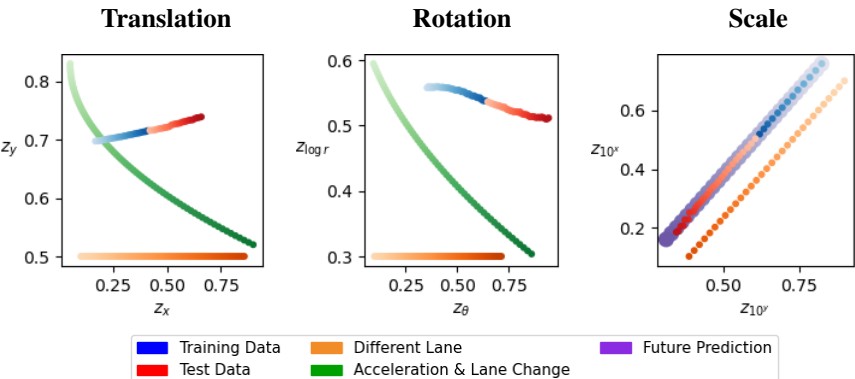

Figure 7: Latent space showing the training data (blue), test data (red), and two different counterfactuals for each experiment (orange, green, purple) that are out of the training distribution. The arrow of time is denoted by the colour intensity, from light to dark.

space in logarithmic coordinates, showing an approximately exponential relationship between the car's position and scale. When compared to the estimated ground truth variables describing the car's position, the latent variables' mean squared error on the test set is an order of magnitude smaller than the best baseline (table 4, MSE, bottom row), which is to be expected as none of the baselines exhibit scale equivariance. Although the reconstruction accuracy for our method is very good (table 4, Acc., bottom row), the CNN and keypoint CNN baseline achieve a slightly higher accuracy. However, for this experiment the reconstruction accuracy is a slightly misleading metric as the cars in the test set are very small, which means the baselines erroneously predicting background where the car should be achieves a better accuracy than our method, which predicts a car at the correct position but imperfectly rendered. Therefore, although our method achieves a very slightly lower reconstruction accuracy than the baselines, the latent mean squared error is still much better than the baselines, showing that the scale equivariance allows the network to generalise successfully to the test set.

**Counterfactual results.** Once the encoder and decoder have been learned, we can use the scale equivariance property of the network to intervene on the latent variables and generate videos of counterfactual scenarios that were never observed at training time. For example, one can extrapolate the latent variables from the training set into the future (fig. 7, right plot, purple colour) to visualise where the car will be in the future assuming constant speed (fig. 8, middle bottom plot), or one can shift the latent variables diagonally (fig. 7, right plot, orange colour) to visualise a car driving in a different lane (fig. 8, bottom right plot). Because none of these scenarios were observed at training time this demonstrates that the model successfully generalises out of the training distribution.

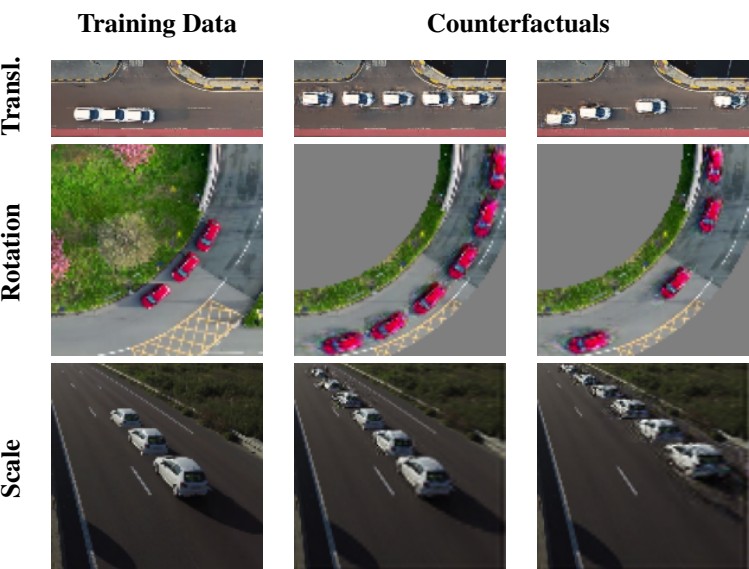

Figure 8: Training data (left column) and two different counterfactuals (middle and right columns) for each experiment (rows). The counterfactuals are obtained by intervening on and decoding the latent variables to obtain out-of-distribution data never seen during training.

