# OpenReview forum: "Identifiable Representation Learning via Architecture Equivariances"
_ICLR.cc/2024/Conference — ICLR 2024 Conference Withdrawn Submission_

### Official Review · Reviewer_MCV9 · 2023-10-16

**Soundness:** 2 fair
**Presentation:** 3 good
**Contribution:** 2 fair
**Rating:** 5
**Confidence:** 4

**Summary:**

This paper proposes an architecture for transformation-equivariant representation learning. Specifically, the
encoder and decoder consist of blocks that make the latent representation equivariant to a specific transformation (i.e.,  scaling, rotation, and coloring). This transformation and its inverse are defined via warping grids that can encode equivariance. Experiments on conducted on real-world video sequences. The authors also show the latent representation is identifiable under certain assumptions.

**Strengths:**

1. The concept of generalized or transformation-specific equivariance is new and interesting. It avoids the limitation of strict group theory and can be applied to general transformations that happen in real-world videos, which allows for more flexibility and diverse application scenarios.

2. The identifiability analysis is sufficient to show the true latent variables can be learned under some assumptions.

3. The counterfactual results demonstrate that the model generalizes well to OoD samples.

**Weaknesses:**

1. **Literature of generalized equivariance**. There are also some recent papers talking about generalized equivariance in which they also define the concept of equivariance as transformation-specific, such as [1,2,3,4]. In particular, [1,3] also employs an autoencoder framework for equivariant representation learning but has a fully probabilistic treatment. These papers share some similarities with this work. I would suggest comparing them and discussing the key differences.

2. **Missing ablation studies**. There are no any sort of ablation studies in the current version of the paper, but there are quite a few components that are worth further exploring. For instance, is the $\texttt{softargmax}$ really necessary to obtain a latent variable, and are there any other alternatives that can substitute Gaussian rendering?

3. **No actual baselines**. One major concern to me is that there is no actual baseline for quantitative comparisons so I have no idea of the performance of the model. The compared baselines are almost like some ablation studies of the proposed model. I would suggest the authors consider adding some baseline of equivariant representation learning.

4. **Role of Softmax**. I did not really understand the role of the softmax operator. It seems to me that the softmax operator is just used for permutation equivariance. However, given that the encoders and decoders are not equivariant, it does not make much sense to me to make the bottleneck part equivariant. In my opinion, the authors might just define a normal latent space instead, which should give more flexibility to learn equivariant representations. Also, the $\texttt{softargmax}$ operator only gives the highest probability of one grid, which should correspond to the object in the video. Then what about multi-object scenarios? Does it make more sense to use a $\texttt{sigmoid}$ gate in this case?

5. **Real-world transformations**. As the authors mentioned in the conclusion, they plan to expand the transformation classes in future work. I would suggest the authors consider some real-world datasets of complex transformations, such as Falcorl3D and Isaac3D. These datasets usually serve as disentanglement datasets but they may also be used for equivariant representation learning [3]. There are also some motion/animation datasets worth a try, such as Tai-chi, fashionvideos, and Mgif.

>[1] Topographic VAEs learn Equivariant Capsules. NeurIPS 2021.
>
>[2] Latent Traversals in Generative Models as Potential Flows. ICML 2023.
>
>[3] Flow Factorized Representation Learning. NeurIPS 2023.
>
>[4] Steerable Equivariant Representation Learning. Arxiv.

**Questions:**

1. Currently the authors use 2D maps as latent variables. How about just using 1-D latent codes and some operators to move the latent samples around?

---

### Official Review · Reviewer_ad2P · 2023-10-27

**Soundness:** 1 poor
**Presentation:** 1 poor
**Contribution:** 1 poor
**Rating:** 3
**Confidence:** 4

**Summary:**

This work proposes a way to learn the position of moving objects in a static camera setting, and then alter the position of those objects in a controllable fashion. The authors do so by employing an auto encoder architecture where the input image is pre-processed (and post-processed on the other end) by an invertible warping function. Such warping function is derived analytically for each type of transformation studied (translation, rotation and scale). The bottleneck representations (latent variables) correspond to the (x,y) coordinates of the moving objects in the image. The value of the latent variables can then be changed so that the decoder can be used to generate new unseen images called "counterfactual" examples. The methods learns to detect the moving objets by taking advantage of multiple frames where the background is fixed and one object is moving so that the transformation of the object can be described as a translation or a rotation or a scale. The authors show that the recovered position is certifiable up to small shifts. The experimental section shows that given short videos (from 11 to 73 frames) the model learn the position of the objects with very small MSE and that those objects can be moved within the same background.

**Strengths:**

I find the ability to generate new unseen but controlled images very compelling for many potential applications.

**Weaknesses:**

The method is limited in a few ways:
- It can only deal with translation, rotation and scale transformation.
- It can only deal (or results are only shown) with one transformation at the time.
- The method requires static background.
- The method requires to be trained for every scenario. This limitation is understandable in some settings (such as security camera) however it is unclear how new objects (not seen during training) would be reconstructed (even if the scenario is fixed).

The method tackles a problem that seems to have been solved by many previous approaches and it is not applicable to more interesting and challenging transformations (such as color transformations), or combination of transformations. Steerable filters (e.g., H-Net, SESN, Deformation robust roto-scale-translation equivariant CNNs, etc...), Group Convolutions..., and other techniques such as Scalable Fourier-Argand Representations, Duet or Equivariant SSL, are already equivariant models with respect to translation, rotation and scale. Even when they focus on classification problems, a decoder can easily be implemented for generation purposes (such as in DUET). Many of these techniques have much less constraints and they can be used to learn (quasi)equivariance representations also with respect to other transformations.

I find some claims to be too vague or not well supported by evidence. For example:
- The claim that the model can generalize to out of training distribution is very broad compared to the results shown. Specifically, the out of distribution capability seems limited to the position of the objects in the scene. It does not extend to new objects or new scenes (even when the transformation is exactly the same).
- From the abstract it reads as the method tackle the problem of "robustness" but it is unclear what this means. Additionally there is no experiment where any robustness aspect is explored.
- Another claim is that representations learned with the proposed method are interpretable. However, with the exception of the (x,y) regressed values there is no other interpretation shown in the experiments.
- The authors show that thanks to the proposed technique one can generate new images (in the paper called counterfactual examples) where the background is kept constant while the position of the main object is changed. However, it is unclear if the quality of the generated examples is "good enough" as there is no experiment that leverages those counterfactual images.

Another weakness in my opinion is the combination of the many and strong assumptions required (See Theorem 1, especially fixed background in all images which greatly reduces the applications) paired with the similar results compared to the Keypoint CNN baseline (Table 3). The proposed model has a lower MSE but the accuracy is comparable. It is unclear if the lower MSE is better in practice. It seems that without any adaptation (required by the proposed method since the warping function needs to be selected based on the specific transformation) and with far less assumptions, Keypoint CCN is a more flexible solution.

The experimental section should be strengthened.
- It is unclear if the results are statistically significant as no error bars are shown.
- Experiments are based on 3 toy datasets with as little as 11 frames (at most 79 frames).
- In table 3 the authors report "Accuracy" but it is unclear how this has been computed given the tasks are about reconstruction or regression.
- In the counterfactual results for scale the manuscript reads "one can extrapolate the latent variables" however it seems this might be an overstatement as the results show some level of generalization albeit limited to interpolation (all scales in between can be reconstructed but during training both extremes seem visible?)
- I do not fully understand how the positional encoding is used. What is its role? How would the model work without it?
- The static background is a strong assumption and no experiment shown how the model is robust to small changes of the background that would be naturally occurring (e.g., lighting conditions or noise)

**Questions:**

I am afraid that the amount of clarifications and changes required to change my opinion would be difficult to address during a rebuttal stage. However, here are some questions (addressing this in the manuscript could also improve its clarity) and suggestions:

- In table 3 the authors report "Accuracy" but it is unclear how this has been computed given the tasks are about reconstruction or regression. This should be explained in the experimental section.
- How can one use those counterfactual images? I would think about ways of showing how these images can be employed in practice. This would provide an application for the method and also an empirical way to show that the quality of the generation is sufficient for such application.
- What is the intuition behind the fact that CNN have a very large MSE compared to all other techniques (order of magnitude larger) but the accuracy is still very good (above 92%)? This could be a useful discussion point for the paper.
- Beyond knowing where the object is (for which one could also use object detection in general, for this simplified case with fixed background the computer vision literature is also providing various method for background subtraction), in which way are these representations interpretable? It seems to me that these representation are "trivially" interpretable in that the objective is to regress the position of a moving object. Perhaps the authors could provide more explanation about what they mean by interpretable and extend the interpretability beyond the object position?
- I would also think how to lift two big restrictions: one is about the static camera, the second is about the limited set of transformations that can be used.
- What is the role of the positional encoding? It is not clearly explained and it would be useful to see an ablation study where positional encoding is not used.
- In Figure 1: why the image is also rotated when warped? Why should the background of the image be reconstructed simply but the (x,y) coordinate? Why the top left and bottom right corner are masked in the reconstruction? They don't seem to have been cut off by the warping function.

---

### Official Review · Reviewer_zkMx · 2023-11-01

**Soundness:** 2 fair
**Presentation:** 2 fair
**Contribution:** 1 poor
**Rating:** 1
**Confidence:** 4

**Summary:**

This paper proposes equivariant auto-encoder architecture that can give identifiable latent representation after training. The main idea is to lift up the translation equivariance of CNN and make it into rotation and scale equivariance by applying certain warping functions. In particular, polar-coordinate warping and logarithmic warping are suggested to be taken before and after the forward of CNN, which makes the architecture equivariant to rotation and scale, respectively. Applying this technique to an auto-encoder, along with the scalarization method also suggested in the paper, enables learning identifiable and equivariant latents. Experiments are conducted on simple surveillance-camera images (fixed background with moving objects), and the proposed method show superior performance compared to the baselines, vanilla MLPs and CNNs.

**Strengths:**

The considered topic is important in most of the computer vision problems. While many image recognition and generation models are showing human-level performance, they often lack generalizability over transformation. If the proposed method works in general, many existing models will benefit from it by large.

**Weaknesses:**

The main weakness of the paper is that the suggested techniques seem to be only applicable in a confined problem setting.

In rotation equivariance scenario, for example, one has to know in advance the pivot point that the objects rotate on. This means we have to heavily rely on the domain knowledge to use the proposed method. Also, the method does not scale with multiple objects, each rotating with its own pivot. Similar goes for the scale equivariance, which requires the knowledge of vanishing point.

Also, the warping seems to make a great distortion. In the rotation scenario, for example, if one car is locating at a much larger radius than the other car, the warped image would make the first car much smaller than the other in size. Then, it becomes a scale problem again.

Another weak point is a missing reference, Alias-Free GANs (Karras et al., 2021). This reference  proposes translation and rotation equivariant architecture based on the discrete-signal-processing theory, which forms a strong baseline for this paper. A proper comparison in experiments seem to be needed.


Minor points:

- Using binary cross-entropy for reconstruction loss of natural images feels like a bad practice. L2 loss (or similar) should be considered first since the pixel values are not binary. In the same sense, reporting accuracy for reconstruction error feels awkward.

**Questions:**

* Can a single network be equivariant to different transformation at the same time? For example, can a rotation-and-scale-equivariance network be built?
* Given there are multiple objects, How is the MSE loss to the ground-truth physical variables measured? How is the permutation handled?

---

### Official Review · Reviewer_rWrK · 2023-11-01

**Soundness:** 2 fair
**Presentation:** 3 good
**Contribution:** 2 fair
**Rating:** 5
**Confidence:** 4

**Summary:**

This paper introduces a convolutional autoencoder along with specialized warp/unwarp and latent variable functions to map input transformations (translation, scale, rotation) to latent space variables. The resulting experiments show that the model can learn to map from input space transformation to latent space, and one can perform interventions in the latent space to enable counterfactual investigations.

**Strengths:**

The topic of equivariance is an important one, and is not very well explored in the literature. The presented method is simple and probably easy to replicate. The ideas are presented in a clear manner. Experimental and quantitative evidence of the counterfactual intervention are provided.

**Weaknesses:**

1. I don't understand the claim that "delta is a pseudo-inverse of the argmax" for a scalar z (defined at the bottom of page 5). This seems trivial for a scalar.

2. Theorem 1 has very restrictive conditions e.g. all images have the same background etc. And in practice, as far as I understand, the experimental results show that each model has to be trained on a particular input video, and the latent variables are specialized to a particular transformation and input sample. Is this correct? If so, this is a very restrictive and artificial setting. Have the authors tried to test the generalization properties of the resulting model to different videos in testing? It feels that the slightest camera motion in the video would cause problems with the approach...

3. How can multiple equivariance properties be combined in one model? This is unclear from the results.

**Questions:**

1. How can this framework support equivariances for properties such as illumination or out of plane rotations?

2. This framework is very specific to CNN's. Any thoughts on how we can make achieve equivariance properties on Transformer architectures which do not use convolutions?